# VO_*x*_ Phase Mixture of Reduced Single Crystalline V_2_O_5_: VO_2_ Resistive Switching

**DOI:** 10.3390/ma15217652

**Published:** 2022-10-31

**Authors:** Brian Walls, Oisín Murtagh, Sergey I. Bozhko, Andrei Ionov, Andrey A. Mazilkin, Daragh Mullarkey, Ainur Zhussupbekova, Dmitry A. Shulyatev, Kuanysh Zhussupbekov, Nikolai Andreev, Nataliya Tabachkova, Igor V. Shvets

**Affiliations:** 1School of Physics and Centre for Research on Adaptive Nanostructures and Nanodevices (CRANN), Trinity College Dublin, D02PD91 Dublin, Ireland; 2Institute of Solid State Physics, Russian Academy of Sciences, 142432 Chernogolovka, Russia; 3Faculty of Physics, Higher School of Economics University, Myasnitskaya Ulitsa, 20, 101000 Moscow, Russia; 4Materials Modeling and Development Laboratory, National University of Science and Technology MISIS, Leninskii Pr. 4, 119991 Moscow, Russia; 5Prokhorov General Physics Institute, Russian Academy of Sciences, Vavilov Str. 38, 119991 Moscow, Russia

**Keywords:** metal oxide, vanadium dioxide, vanadium pentoxide, resistive switching, phase transition, single crystalline, oxide reduction

## Abstract

The strongly correlated electron material, vanadium dioxide (VO${}_{2}$), has seen considerable attention and research application in metal-oxide electronics due to its metal-to-insulator transition close to room temperature. Vacuum annealing a V${}_{2}$O${}_{5}$(010) single crystal results in Wadsley phases (V${}_{n}$O${}_{2n+1}$, *n* > 1) and VO${}_{2}$. The resistance changes by a factor of 20 at 342 K, corresponding to the metal-to-insulator phase transition of VO${}_{2}$. Macroscopic voltage-current measurements with a probe separation on the millimetre scale result in Joule heating-induced resistive switching at extremely low voltages of under a volt. This can reduce the hysteresis and facilitate low temperature operation of VO${}_{2}$ devices, of potential benefit for switching speed and device stability. This is correlated to the low resistance of the system at temperatures below the transition. High-resolution transmission electron microscopy measurements reveal a complex structural relationship between V${}_{2}$O${}_{5}$, VO${}_{2}$ and V${}_{6}$O${}_{13}$ crystallites. Percolation paths incorporating both VO${}_{2}$ and metallic V${}_{6}$O${}_{13}$ are revealed, which can reduce the resistance below the transition and result in exceptionally low voltage resistive switching.

## 1. Introduction

The metal-insulator transition (MIT) of VO${}_{2}$ and a variety of other compounds is of long-standing interest due to both the rich physics at play and its application in electronics such as oscillators, neuromorphic and memory devices [1,2].

Across 0.1 K at 339 K, single crystalline VO${}_{2}$ undergoes a structural transition accompanied by a 5 order of magnitude change in conductivity [3]. There is an intrinsic thermal hysteresis of ∼1 K [3]. The phase change can be induced electrically and is termed resistive switching (RS). It is clear that Joule heating [4,5,6,7,8] or an electric field [9] can induce RS. Isolating one mechanism is challenging and it appears that even when one mechanism dominates, the other cannot be ignored. This is especially the case for the electric field dominated switching where the associated current can induce Joule heating [10].

Characteristics of the MIT are influenced by stoichiometry [11,12], crystallinity [13] and the substrate in the case of thin films [14]. Inhomogeneity in the stoichiometry widens the transition due to the range of different transition temperatures. Polycrystalline thin films, with a range of stoichiometries due to grain boundaries, exhibit a wider transition than single crystals [14]. The MIT temperature (T${}_{\mathrm{MIT}}$) is also very sensitive to substrate-induced strain [15,16].

The vanadium-oxygen system is complex, consisting of several single valence phases (2+ to 5+) and mixed valence Magnéli (3+ and 4+) and Wadsley (4+ and 5+) phases. Many Magnéli phases and V${}_{6}$O${}_{13}$ exhibit MITs at temperatures ranging from 70 K (V${}_{8}$O${}_{15}$) to 430 K (V${}_{3}$O${}_{5}$) [17]. The thermal reduction of single crystalline V${}_{2}$O${}_{5}$ at around 800 K in high vacuum [18] and oxygen [19] environments result in conversion to V${}_{6}$O${}_{13}$. The reduction of thin films are in contrast to single crystals; Ramana et al. converted amorphous V${}_{2}$O${}_{5}$ to V${}_{2}$O${}_{3}$ at 860 K [20], while Monfort et al. converted polycrystalline V${}_{2}$O${}_{5}$ to V${}_{4}$O${}_{9}$ at 673 K and to VO${}_{2}$(B) at 773 K [21]. Evidently the nature of the V${}_{2}$O${}_{5}$ determines the reduction, which can be due to crystallinity, defects or both.

The electric field-induced transition requires fields on the order of 10${}^{7}$ V/m [9]. In Joule heating-dominated RS, the electric field is a few orders of magnitude smaller [4,5,6,7,8]. Reducing the switching electric field, or switching voltage, can reduce the power surge at the transition point limiting additional heating, and therefore improving switching speed, in principle. Furthermore, the low field can facilitate lower device operating temperatures, which can be optimal for switching speed [22] and device stability. Finally, it can reduce the electronic hysteresis [23]. The ideal hysteresis profile varies between application [2].

In this work, thermal high vacuum (HV) reduction of a V${}_{2}$O${}_{5}$ single crystal results in several VO${}_{x}$ phases in the surface region including V${}_{2}$O${}_{5}$, V${}_{6}$O${}_{13}$ and VO${}_{2}$. Joule heating-induced resistive switching of the VO${}_{2}$ is realised at exceptionally low voltages. This is due to the low resistance of the system below the $T_{MIT}$ correlated to the structure of the phases in the surface region. High-resolution transmission electron microscopy (TEM) reveals a complex structural relationship between VO${}_{2}$, V${}_{6}$O${}_{13}$ and V${}_{2}$O${}_{5}$. Percolation paths—incorporating *both* VO${}_{2}$ and metallic V${}_{6}$O${}_{13}$— can reduce the resistance at temperatures below the $T_{MIT}$ and result in the phenomenally low voltage resistive switching. The use of a single crystal facilitates the examination of highly ordered, low defect density V${}_{2}$O${}_{5}$. As we have established, the crystallinity influences the reduction process, and therefore will influence the structural aspects of the phase mixture and the correlated switching properties of the VO${}_{2}$ therein.

The paper is structured as follows; (i) the experimental details are outlined, (ii) we characterise the reduction of the V${}_{2}$O${}_{5}$(010), focusing on one particular reducing condition which produces the low voltage resistive switching of VO${}_{2}$, (iii) macroscopic 4-point probe electrical measurements inducing Joule heating resistive switching are presented and the comparison to the literature is discussed and (iv) finally, TEM measurements visualise the phases in the surface region, which informs the mechanism of the low voltage resistive switching.

## 2. Materials and Methods

V${}_{2}$O${}_{5}$ single crystals have been grown by the floating zone technique. The growth process was carried out in oxygen flow at a crystallisation rate 8 mm/h. Cleavage of the V${}_{2}$O${}_{5}$ single crystal is achieved easily and reveals a reflective V${}_{2}$O${}_{5}$(010) surface. The crystal size are in the range of 10 × 2 × 1 mm. A cleaved crystal is annealed in a vacuum pressure of 10${}^{-5}$ mbar with the objective of reducing the crystal to form VO${}_{2}$. Electrical and x-ray diffraction (XRD) measurements were performed before and after the thermal treatment to investigate the change in conductivity and phases present. Voltage-driven voltage-current (VI) measurements have been performed on the annealed crystal, which contains VO${}_{2}$ in the surface region revealed by electrical measurements and XRD. The VI measurements were performed in a narrow temperature window around the VO${}_{2}$
$T_{MIT}$ in a rough vacuum pressure of 10 mbar. The thermocouple and crystal are placed in close proximity on a sapphire plate, which sits on top of a heater. The VI curves are measured by the linear four-point probe method. The probes contact the (010) surface and are separated by 2 mm. XRD patterns have been measured with a Bruker D8 Discover using a Cu-Kα source. X-ray photoelectron spectroscopy (XPS) measurements have been performed using an Omicron MultiProbe XPS system using monochromated Al Kα X-rays (XM 1000, 1486.7 eV) with an instrumental resolution of 0.6 eV. Spectra have been analysed and fitted with CasaXPS. For the TEM investigation a different sample from the same as-grown crystal was annealed in the same conditions prior to lamella preparation. The samples for the TEM investigations were prepared on VERSA 3D HighVac dual beam facility (FEI) using the Ga ion beam. Regular focused ion beam lamella preparation was carried out at 30 kV and followed by 2 kV final step to improve the surface quality. TEM investigations were performed on Titan 80-300 (FEI) TEM equipped with image corrector and JEM 2100 transmission electron microscope.

## 3. Results

XRD measurements of the pristine crystal are depicted in Figure 1. The θ–2θ scan reveals sharp, pronounced V${}_{2}$O${}_{5}$<010> peaks [24]. Additional peaks, highlighted by asterisks, are due to the sample stage. The reciprocal space map (RSM) of the V${}_{2}$O${}_{5}$(010) peak is characteristic of a single crystal (Figure 1b). The distribution in ω indicates there is a small angular distribution of crystallites about the surface normal. This is best visualised by the two peaks. The sharpness in 2θ indicates the crystallites are large with a consistent lattice parameter. The lattice constant (4.354 Å) closely matches that of single crystalline V${}_{2}$O${}_{5}$ [24].

It is noted that the reduction is strongly dependent on the annealing temperature; At 600 K V${}_{2}$O${}_{5}$ and V${}_{4}$O${}_{9}$ is observed; at 700 K and 800 K V${}_{2}$O${}_{5}$, V${}_{4}$O${}_{9}$, V${}_{6}$O${}_{13}$, VO${}_{2}$(B) and VO${}_{2}$ is observed; at 900 K V${}_{2}$O${}_{5}$, V${}_{6}$O${}_{13}$, VO${}_{2}$(B) and VO${}_{2}$ is observed with V${}_{6}$O${}_{13}$ and V${}_{2}$O${}_{5}$ dominating. The resistance and XRD measurements for different annealing temperatures are presented in Appendix A. In the presented work we focus on the conditions which produce the most pronounced VO${}_{2}$ contribution and low voltage resistive switching. This was realised by annealing for 16 h at 800 K in a pressure of 10${}^{-5}$ mbar. The measurements presented henceforth are performed on the crystal annealed under these conditions. XRD measurements of the crystal after vacuum annealing for 16 h at 800 K in a pressure of 10${}^{-5}$ mbar are depicted in Figure 2.

Due to the multitude of phases and their low-symmetry nature, there are a vast amount of vanadium oxide diffraction peaks. In addition, the crystal structure of some of these phases is not well understood, certainly related to the difficulty in synthesising mixed valence phases as a single crystalline product. The Wadsley phases of the vandadium-oxygen system (V${}_{n}$O${}_{2n+1}$) range from V${}_{2}$O${}_{5}$ (*n* = 2) to VO${}_{2}$(B) (*n* = *∞*). Other Wadsley phases include V${}_{6}$O${}_{13}$ and V${}_{4}$O${}_{9}$. V${}_{2}$O${}_{5}$ is orthorhombic and consists of a V${}_{2}$O${}_{3}$-O${}_{2}$ stacking sequence along its [010] direction. Orthorhombic V${}_{4}$O${}_{9}$ consists of a V${}_{2}$O${}_{3}$-O${}_{2}$-V${}_{2}$O${}_{3}$-O stacking sequence along its [001] direction [25] and is based on a sublattice of oxygen vacancies within V${}_{2}$O${}_{5}$ residing in the oxygen layers [26]. Reports of V${}_{4}$O${}_{9}$ are scarce; it is synthesized via the reduction of V${}_{2}$O${}_{5}$[21,25,26,27,28] with V${}_{4}$O${}_{9}$[001] aligned to V${}_{2}$O${}_{5}$[010] [21,25,26]. Along its [001] direction V${}_{6}$O${}_{13}$ consists of a V${}_{2}$O${}_{3}$-O${}_{2}$-V${}_{2}$O${}_{3}$-V${}_{2}$O${}_{3}$-O${}_{2}$ stacking sequence. V${}_{6}$O${}_{13}$ can be formed from V${}_{2}$O${}_{5}$ via the removal of every third O${}_{2}$ plane. This reduction and a shear operation produces monoclinic V${}_{6}$O${}_{13}$ from orthorhombic V${}_{2}$O${}_{5}$[18]. Similarly to V${}_{2}$O${}_{5}$ and V${}_{6}$O${}_{13}$, VO${}_{2}$(B) consists of V${}_{2}$O${}_{3}$ and O${}_{2}$ planes. The stacking sequence is V${}_{2}$O${}_{3}$-V${}_{2}$O${}_{3}$-O${}_{2}$. This corresponds to removing every second O${}_{2}$ plane of V${}_{2}$O${}_{5}$. This combined with a shear operation can produce monoclinic VO${}_{2}$(B). This results in VO${}_{2}$(B)[001] aligned with V${}_{2}$O${}_{5}$[010]. Members of the Wadsley (V${}_{n}$O${}_{2n+1}$) series cannot be related to Magneli (V${}_{n}$O${}_{2n-1}$) phases by a common substructure. As such, relating the crystal structure of VO${}_{2}$ to VO${}_{2}$(B) or the parent V${}_{2}$O${}_{5}$ is impractical. This is emphasised by the complex transformation of VO${}_{2}$(B) to VO${}_{2}$; it involves an initial disordering step followed by a reordering into VO${}_{2}$[29,30]. The (110) surface of tetragonal VO${}_{2}$—corresponding to the (011) surface of the room temperature monoclinic phase—is predicted to be energetically favourable [31] and it was observed to form surfaces of crystalline VO${}_{2}$ nanoparticles [32,33]. Returning to the XRD of the annealed crystal, the additional peaks in the θ–2θ scan are attributed to VO${}_{2}$<110> and <011> [34], V${}_{6}$O${}_{13}$<001>, <010> and <102> [35] and V${}_{4}$O${}_{9}$<001> [25]. The entire scan, ranging from 10–100°, is presented in Appendix A. V${}_{6}$O${}_{13}$<001> and V${}_{4}$O${}_{9}$<001> are in-line with the discussed reduction mechanisms and literature, V${}_{6}$O${}_{13}$<010> is confirmed by TEM measurements presented later, V${}_{6}$O${}_{13}$<012> is assigned tentatively and VO${}_{2}$ is confirmed by resistance measurements. We note that in contrast to the work of Colpaert et al. [18], who annealed a V${}_{2}$O${}_{5}$(010) single crystal under similar conditions and reported V${}_{6}$O${}_{13}$ and V${}_{2}$O${}_{5}$, in this work we observe additional phases, most notably VO${}_{2}$.

The reciprocal space maps of the V${}_{2}$O${}_{5}$(010) and VO${}_{2}$(011) peaks are presented in Figure 2b,c. Modification of the V${}_{2}$O${}_{5}$(010) region in comparison to the pristine crystal is evident; the RSM is characterised by two sharp components and a homogeneously distributed component. The peaks correspond to 4.415 Å and 4.409 Å while the band corresponds to 4.388 Å. All of these values are greater than the pristine lattice constant. This is in agreement with the work of Monfort et al. A shift of the V${}_{2}$O${}_{5}$<010> diffraction peaks to smaller angles and the simultaneous formation of VO${}_{2}$ and V${}_{4}$O${}_{9}$ after vacuum annealing a V${}_{2}$O${}_{5}$ thin film was observed by [21]. The intensity of the VO${}_{2}$(011) peak is homogeneous indicating the orientation of VO${}_{2}$ crystallites is distributed evenly within the probed angular window. The (011) inter-planar distance of 3.22 Å is slightly smaller than the literature value of 3.27 Å [34].

The width of the reflexes in the 2θ direction are analysed in Figure 2d–f. The width can be due to variation in the lattice constant and/or coherent domain size. Applying the FWHM of the VO${}_{2}$(011) reflex to the Scherrer equation gives an average coherent domain size of 34 nm. The instrumental broadening has been estimated by measuring the FWHM of a single crystal silicon (111) peak at 2θ = 28.44°. If one interprets the width to be due to variation of the lattice constant we obtain a maximum deviation from the pristine value of around 0.3 Å, or 1%. However, strain is known to substantially alter the $T_{MIT}$: Cao et al. demonstrate a 15 K shift per percent strain. Considering the sharp transition observed in resistance measurements in Figure 3 discussed below, variation in the lattice constant is excluded and we estimate the average crystallite size from the FWHM of the VO${}_{2}$ reflex in the 2θ direction. The V${}_{2}$O${}_{5}$(010) reflex of the annealed crystal gives a coherent domain size in the region of 100–200 nm, although variation in the lattice constant cannot be excluded. In the case of the pristine crystal the V${}_{2}$O${}_{5}$(010) reflex is too sharp to accurately apply to the Scherrer equation; this is indicative of very large crystallites. Evidently the thermal treatment reduces the crystallinity of the V${}_{2}$O${}_{5}$.

Figure 3 depicts a resistivity measurement of the pristine and annealed crystal in the temperature range 325–380 K with the values normalised to the V${}_{2}$O${}_{5}$ resistivity at 325 K. The resistivity of a pristine crystal at 325 K has been calculated to be (3.0 ± 0.7) × 10${}^{3}$ Ωcm, comparable to [36]. Focusing on the annealed crystal, there is a clear drop in resistance at 342 K characteristic of the VO${}_{2}$ MIT. This transition sees a reduction in the resistance by a factor of 20 across ∼5 K. A thermal hysteresis of 2 K is present, in comparison to the 1 K hysteresis observed in the case of single crystalline VO${}_{2}$[3]. The VO${}_{2}$ is concluded to exhibit only a small range of stoichiometry around dioxide stoichiometry due to the narrow width and hysteresis of the transition.

XPS measurements of the pristine and annealed crystal are depicted in Figure 4. The V 2p region of the pristine crystal only contains a 5+ contribution. The O 1s region of the annealed crystal can be deconvoluted into the dominant peak of lattice oxygen and a peak at higher binding energy corresponding to oxygen vacancies created upon annealing, as has been observed in other metal oxides [37]. The V 2p regions display 5+ and 4+ contributions in line with XRD measurements. The energetic positions of oxygen vacancies, lattice oxygen and V 2p components along with the energy difference between lattice oxygen and each V 2p component—in the case of the pristine and annealed crystal—are in agreement with [38,39]. XRD measurements reveal the overall V 2p line shape is due to several phases including V${}_{2}$O${}_{5}$ (5+), VO${}_{2}$ (4+) and mixed valence V${}_{6}$O${}_{13}$ and V${}_{4}$O${}_{9}$. Therefore, we cannot estimate the proportion of each phase in the surface region.

We now turn to the electrical measurements of this crystal. VI measurements are depicted in Figure 5a. The set temperature ($T_{S}$) has been varied between 337 and 343 K in one degree steps. The voltage is sweeped from 0 → 1 V → 0 → −1 V→ 0. At each set temperature the VI measurement has been performed 6 times with negligible drift seen in the switching characteristics (see Appendix A). In this voltage range RS is observed for $T_{S}$ of 339 K and greater. At the switching threshold voltage ($V_{th}$) the current abruptly increases as the VO${}_{2}$ transitions to the metallic phase. $V_{th}$ decreases as the $T_{S}$ approaches $T_{MIT}$. Considering the electric field in this experiment—500 V/m at 1 V—it is clear the RS can not be due to the electric field: a field of over 10${}^{7}$ V/m is required for electric field-induced resistive switching of VO${}_{2}$[9]. Within the Joule heating picture the power induces heating, some of which is conducted away. If the VO${}_{2}$’s (or a channel therein) temperature surpasses $T_{MIT}$, it will switch to the metallic phase. At this point, the current and power spike can further increase the temperature within the rutile VO${}_{2}$ volume, which carries a large proportion of the current. The net heat added to the system is [6,8]: (1)$$\frac{dQ}{dt}=\frac{V^{2}}{R}-k\left(T_{C}-T_{S}\right)$$

*V* is the voltage drop across 2 mm of the crystal with a resistance *R*. $T_{C}$, $T_{S}$ and *k* are the current carrying channel temperature, set temperature and the effective thermal conductance. If one assumes a steady state, the square of the voltage is proportional to the temperature difference between the channel and the rest of the system. When *V* = $V_{th}$, $T_{C}$ = $T_{MIT}$: (2)$$V_{th}^{2}=Rk\left(T_{MIT}-T_{S}\right)$$ If $T_{S}$ is reduced, additional voltage is required to induce an RS, as is observed. $V_{th}^{2}$ is plotted against $T_{S}$ in Figure 5b. $V_{th}^{2}$ is taken as the maximum of the second derivative. The linearity is testament to the Joule heating mechanism. We assume that *R* and *k* are constant in the temperature range.

When VO${}_{2}$ is in the rutile state and the voltage is sweeping down, the VO${}_{2}$ switches back to the monoclinic state at a voltage lower than $V_{th}$. Clearly the power at any given voltage is greater in the low-resistance state. The power when the material switches to the metallic state will be approximately equal to the power when it switches back to the insulating state, the only variance is due to the temperature hysteresis (see Figure 3a). The sharpness of the RS is proportional to $V_{th}$. The power increases more dramatically when the voltage is greater, giving rise to more rapid heating. Studies which report considerably larger threshold voltages report an extremely sharp switch [5,6].

To the best of the author’s knowledge the threshold voltage per unit length is orders of magnitude lower than any reports in the literature. The reproducibility is discussed is the SI. In Table 1 studies reporting Joule heating-induced resistive switching of VO${}_{2}$ are compared. The power per Kelvin required to induce the RS is comparable to other work performed with a VO${}_{2}$ single crystal in which Joule heating is concluded to be the dominant mechanism [6]. The low threshold voltage is correlated to the low resistance of the system at temperatures below $T_{MIT}$.

In order to understand the mechanism of the exceptionally low resistive switching voltage, high-resolution TEM (HRTEM) measurements have been performed. In Figure 6a a TEM overview image of the pristine sample cross-section is shown along with the corresponding diffraction pattern (DP). The upper part of the image corresponds to the surface of the V${}_{2}$O${}_{5}$ sample coated with the protective carbon layer. The image does not show any features in the sample like cracks, grain boundaries or dislocations, i.e., on the scale of the prepared lamella the sample is practically a perfect single crystal. The DP also supports this conclusion. Indexing of the DP shows good coincidence with the V${}_{2}$O${}_{5}$ orthorhombic phase [24] with the surface plane parallel to the [100] crystallographic direction. The schematic in the image shows the orientation of the main V${}_{2}$O${}_{5}$ lattice directions.

The structure of the sample changes dramatically after the annealing at 800 K. A low magnification TEM image in Figure 6b shows that the sample is divided into a number of rectangular areas. While there is contrast in all areas of this annealed sample, the most significant features are 1–2 microns from the surface. The main reflections in the DP, depicted inset, collected from the entire sample correspond again to the V${}_{2}$O${}_{5}$ phase. However, some additional weaker reflections indicated by the blue oval in the DP are also observed. There is not enough of these reflections for unambiguous indexing. This will be done in the proceeding paragraph based on the HRTEM data.

Detailed analysis of the sample structure was performed on a small part of the sample indicated by the red dashed square in Figure 6b. An image of this area is shown in Figure 6c. The colour assignment is the result of the phase analysis based on the HRTEM and nanobeam diffraction data. Red, blue and green correspond to V${}_{2}$O${}_{5}$, V${}_{6}$O${}_{13}$ and VO${}_{2}$ (monoclinic), respectively. Part of the image indicated by arrows contains no material due to the FIB milling. The main part of the sample is occupied by the orthorhombic V${}_{2}$O${}_{5}$ phase which corroborates with the DP from the entire sample (Figure 6b). A fast Fourier transform (FFT) of area 1 is depicted in Figure 6d. All the individual areas occupied by V${}_{2}$O${}_{5}$ have the same orientation in the sample plane and along the normal direction. Another structural feature in the annealed sample (blue) is areas over a micron in size in the vertical direction and 100–200 nm in size in the horizontal direction. The FFT from area 2 is shown in Figure 6e. Indexing of the FFT gives the match with the monoclinic V${}_{6}$O${}_{13}$ structure [35]. All the individual areas of Figure 6c occupied by V${}_{6}$O${}_{13}$ have the same orientation, which is depicted in Figure 6b. The presence of this phase explains several weak reflections in the DP from the annealed sample (see blue oval inset of Figure 6b). The surface layer of the sample consists of large grains (green) with an undefined shape. The nanobeam diffraction technique was employed to analyse their crystallography; According to the data obtained, these grains belong to the VO${}_{2}$ phase [34] and have an arbitrary orientation. The nanobeam DP demonstrated in Figure 6f corresponds to area 3 of the sample.

A remarkable structural feature of the annealed sample is the splitting of the initially single crystalline V${}_{2}$O${}_{5}$ phase by narrow extremely defined strips in the direction parallel to the sample surface. A higher magnification HRTEM image from the area, highlighted by the green dashed square in Figure 6c, is shown in Figure 6g. The FFT inset—corresponding to the strip—reveals it consists of VO${}_{2}$ and an amorphous phase giving rise to the halo in the FFT.

## 4. Discussion

The assigned phases of the annealed crystal are V${}_{4}$O${}_{9}$, V${}_{6}$O${}_{13}$, VO${}_{2}$ and V${}_{2}$O${}_{5}$. Of these, only V${}_{6}$O${}_{13}$ is metallic at temperatures below the $T_{MIT}$. The most intense reflections in the XRD, besides V${}_{2}$O${}_{5}$, are assigned to V${}_{6}$O${}_{13}$ and this phase is observed in the TEM. We conclude that this phase gives rise to the low resistance of the system at temperatures below the $T_{MIT}$.

In the linear electrical measurements the current path is along V${}_{2}$O${}_{5}$[001], corresponding to the horizontal in the TEM images. The TEM clearly reveals that the current will flow predominately in the surface region where the oxygen content and resistance is reduced. A percolative path incorporating *both* metallic V${}_{6}$O${}_{13}$ and semiconducting VO${}_{2}$ can be envisaged. V${}_{6}$O${}_{13}$ can reduce the resistance of the system below the VO${}_{2}$ transition temperature, however the metallic V${}_{6}$O${}_{13}$ will not short the VO${}_{2}$ due to its confined length in the horizontal direction. This percolation path can reduce the threshold voltage for switching of the VO${}_{2}$ crystallites. This is in agreement with the work of Lin et al. [23] who measured and simulated the threshold voltage as a function of the resistance of the system at low temperatures; the threshold for switching reduces with reducing resistance. The resistance of the low-temperature, monoclinic VO${}_{2}$ phase can be reduced with the inclusion of oxygen vacancies [12]. However, this reduces the $T_{MIT}$ and widens the transition. The $T_{MIT}$ and width of the transition strongly suggest that the VO${}_{2}$ in the system is stoichiometric. It is noted that due to the inclusion of the metallic phase, the entirety of the voltage will not drop over the VO${}_{2}$.

The low resistance of the system below the $T_{MIT}$ results in a small resistance change at the transition of 20 in comparison to 5 orders of magnitude for single crystalline VO${}_{2}$[3]. However, the resistance change in VI measurements is also around 20 indicating the vast majority of the VO${}_{2}$ layer switches to the metallic state. In thin film [40] and single crystalline VO${}_{2}$[6] the resistance change seen in VI is often orders of magnitudes lower than the resistance change upon heating. However, a comparable thermal and electronic transition has been observed [41]. The lower transition magnitude in the electronically driven phase change can be understood by conduction channel formation; a portion, or channel, of the VO${}_{2}$ switches metallic when the threshold voltage is applied [5,8]. Yoon et al. visualized a dependence between device length (electrode separation) and conduction channel size by optical microscopy [5]; At L = 40 μm the conduction channel width is around a third of the device width, while at L = 5 μm the conduction channel width is around a few percent of the device width. In this work the electrodes are mm apart, and hence, comparable thermal and electronic transition magnitudes is reasonable. Furthermore, the channel width is proportional to the amount of current [5]. In our work the current is high due to the low resistance.

A similar study of V${}_{2}$O${}_{5}$ thin films can facilitate micron-scale probe width, which—in theory—will reduce the switching voltage [5,23] and can allow low voltage switching at operating temperatures close to room temperature. One should pay particular attention to the crystallinity of the thin film as it can influence the reduction process [18,20,21]. The phase mixture of reduced single crystalline V${}_{2}$O${}_{5}$ presented in this work is novel; to the best of our knowledge VO${}_{2}$ and V${}_{6}$O${}_{13}$ have not been observed concurrently in reduced V${}_{2}$O${}_{5}$ and a spatially resolved examination of the surface region of reduced V${}_{2}$O${}_{5}$ has not been presented. It should be noted that at smaller device lengths the conducting channel width reduces [5], giving rise to a smaller resistance change at the transition.

The reduction of V${}_{2}$O${}_{5}$ is clearly complex, depending on the annealing temperature, as we have demonstrated, but also the nature of the V${}_{2}$O${}_{5}$, whether it be amorphous, polycrystalline or single crystalline [18,20,21]. We have also revealed there exist a complex arrangement of the different phases in the surface region. In-depth knowledge of the reduction dynamics and mechanisms, focusing on the phases but also their relative geometric structure, is invaluable to the optimisation of this system from a resistive switching point of view, but it is also of great interest from a fundamental viewpoint.

## 5. Conclusions

Vacuum annealing of (010) orientated V${}_{2}$O${}_{5}$ at 800 K sees the formation of VO${}_{2}$ and lower oxide Wadsley phases V${}_{4}$O${}_{9}$ and V${}_{6}$O${}_{13}$. Resistance measurements reveal a resistance change of a factor of 20 at 342 K corresponding to the VO${}_{2}$ metal-to-insulator transition. The transition temperature and relatively small transition width and hysteresis suggest the VO${}_{2}$ is stoichiometric and is of good crystalline quality.

Voltage-current measurements induce resistive switching of the VO${}_{2}$ volume. The dependence between the switching voltage and the crystal temperature indicates Joule heating is the resistive switching mechanism. The voltage is extremely low at under a volt for a probe separation of 6 mm and represents exceptionally low volts per distance resistive switching, which can facilitate lower device operating temperatures beneficial for switching speed and device stability. Transmission electron microscopy measurements reveal a complex arrangement of crystallites in the surface region assigned to V${}_{2}$O${}_{5}$, V${}_{6}$O${}_{13}$ and VO${}_{2}$. In the direction of the current in electrical measurements, percolation paths incorporating *both* V${}_{6}$O${}_{13}$ and VO${}_{2}$ can be realised. The inclusion of metallic V${}_{6}$O${}_{13}$ reduces the resistance of the system below $T_{MIT}$ and the correlated threshold voltage for resistive switching.

## Figures and Tables

**Figure 1 materials-15-07652-f001:**
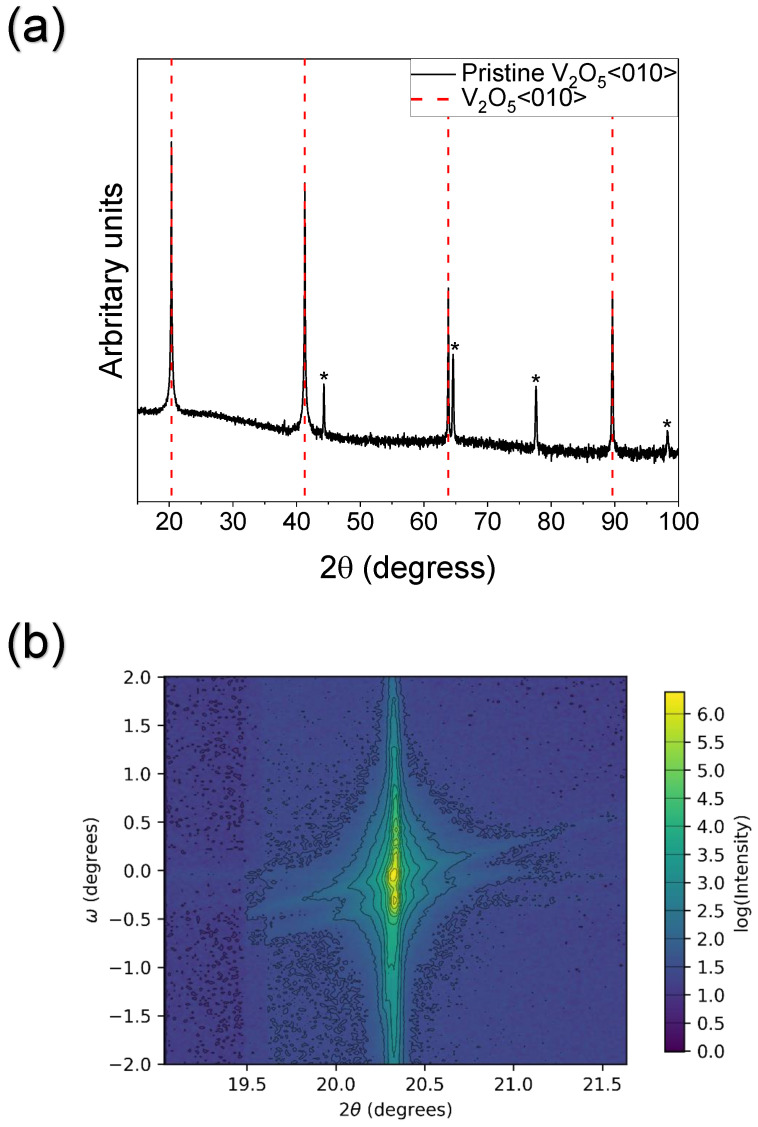
XRD measurements of a pristine, cleaved V${}_{2}$O${}_{5}$(010) single crystal. (**a**) The θ–2θ measurement reveals sharp V${}_{2}$O${}_{5}$<010> peaks. The asterisk is due to the sample stage. (**b**) The reciprocal space map of the V${}_{2}$O${}_{5}$(010) peak is characteristic of a single crystal.

**Figure 2 materials-15-07652-f002:**
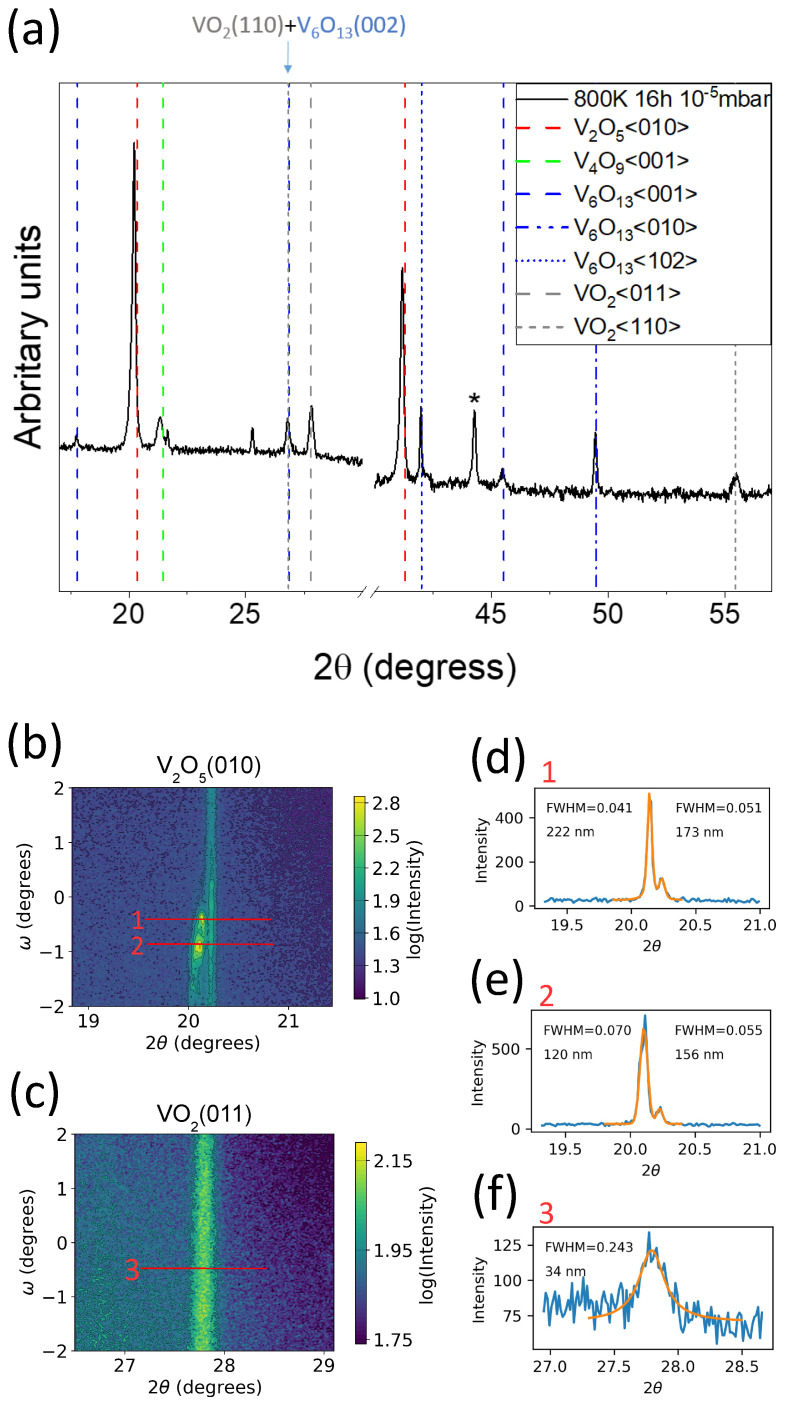
(**a**) θ–2θ measurement of the annealed crystal. Peaks are attributed to V${}_{2}$O${}_{5}$, V${}_{6}$O${}_{13}$, V${}_{4}$O${}_{9}$ and VO${}_{2}$. The asterisk is due to the sample stage. (**b**,**c**) reciprocal space map of the V${}_{2}$O${}_{5}$(010) and VO${}_{2}$(011) peaks of the annealed crystal. V${}_{2}$O${}_{5}$(010) consists of peaks and a homogeneously distributed component at slightly different lattice constants. VO${}_{2}$(011) is homogeneous within the probed angular window. The cross sections of the red line segments 1–3 are depicted in the (**d**–**f**). Applying the full width at half maximum of these peaks to the Scherrer equation gives the average coherent domain size. The crystallites of the VO${}_{2}$ are smaller than that of V${}_{2}$O${}_{5}$.

**Figure 3 materials-15-07652-f003:**
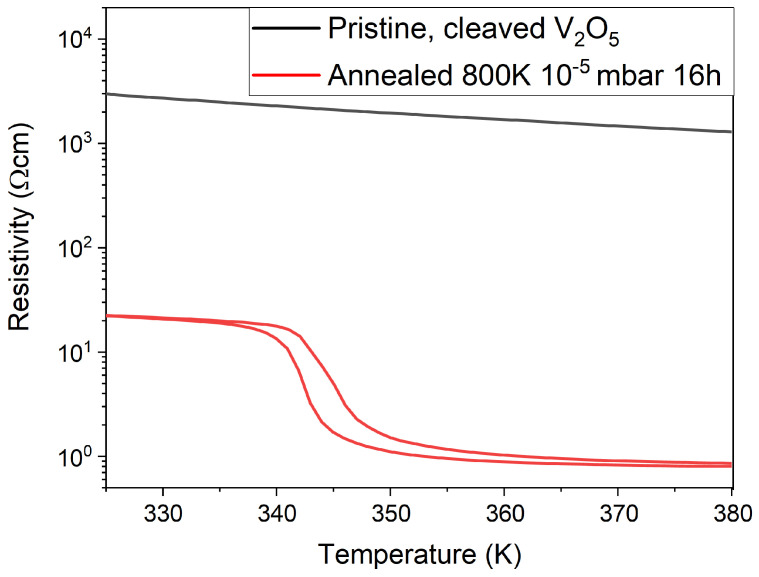
Resistivity as a function of temperature of the pristine crystal and the crystal annealed at 800 K for 16 h at 5 × 10${}^{-5}$ mbar. The VO${}_{2}$ MIT is evident. The data is normalized to the resistivity of V${}_{2}$O${}_{5}$.

**Figure 4 materials-15-07652-f004:**
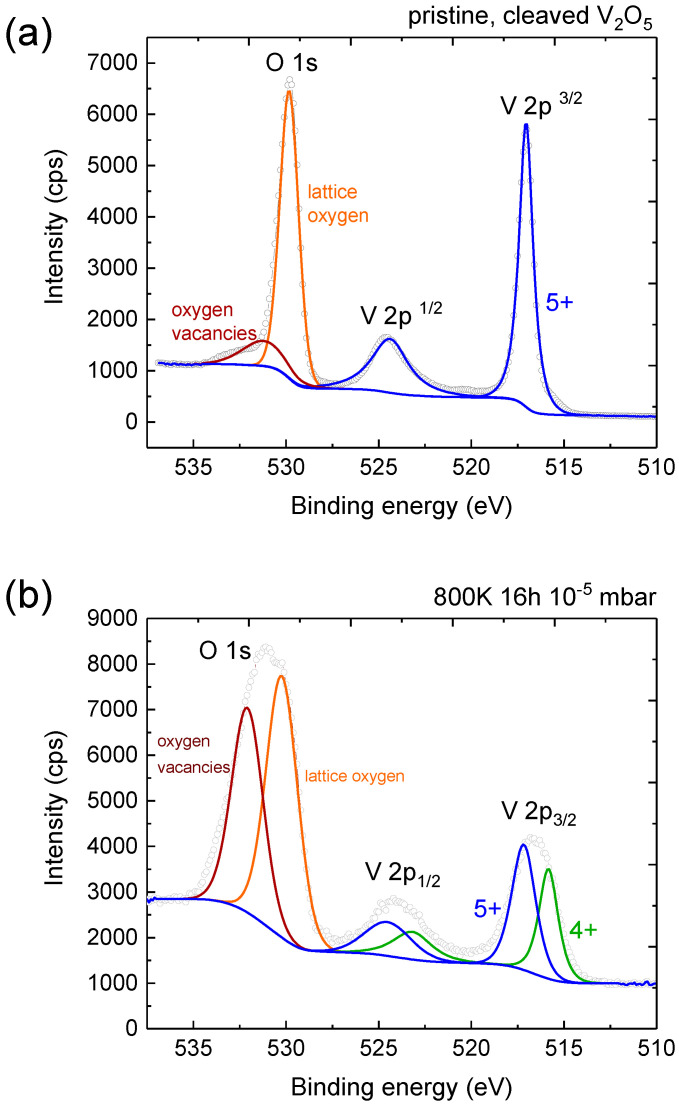
X-ray photoelectron spectra of the cleaved crystal (**a**) and the crystal annealed at 800 K for 16 h in a pressure of 10${}^{-5}$ mbar (**b**). The spectra of the cleaved crystal is characteristic of V${}_{2}$O${}_{5}$. The annealing results in 4+ contribution to the V 2p and an oxygen peak attributed to oxygen vacancies.

**Figure 5 materials-15-07652-f005:**
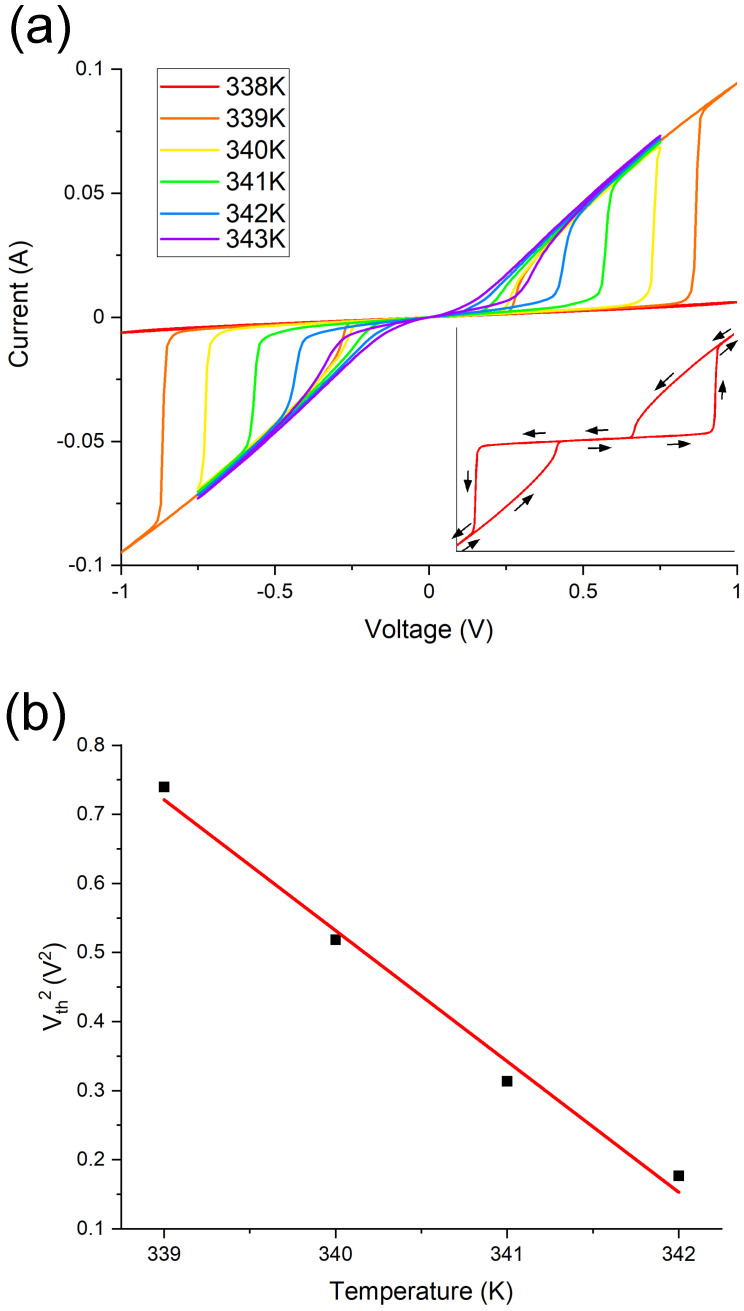
(**a**) Voltage–current measurements at a range of set temperatures. At the threshold voltage the current sharply increases corresponding to a drop in the resistance. This corresponds to the VO${}_{2}$ switching to its metallic state. As the set temperature is reduced the threshold voltage increases. (**b**) plots the square of the threshold voltage against the set temperature. The linearity is evidence for a Joule heating-induced switch.

**Figure 6 materials-15-07652-f006:**
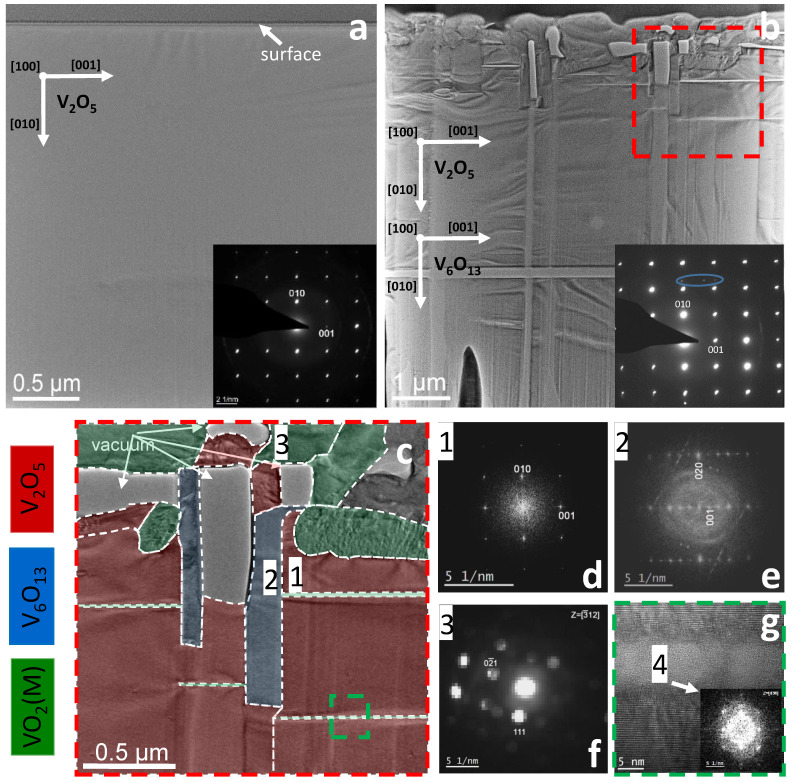
Structure of the vanadium pentoxide crystal in (**a**) as-grown state and (**b**–**g**) after annealing at 800 K for 16 h at 10${}^{-5}$ mbar. (**a**) TEM image of the as-grown sample. (**b**) TEM overview image of the sample after annealing at 800 K. Inserts in (**a**,**b**) show DPs from the entire sample and the orientation of the phases relative to the sample frame. (**c**) enlarged fragment of (**b**) indicated by the red dashed square; the colour of the image areas depends on the phase allocation; the colour code is at the bottom left of the figure. (**d**,**e**) FFTs from the HRTEM images of areas 1 and 2 in (**c**), which are determined to correspond to orthorhombic V${}_{2}$O${}_{5}$ in 1 and monoclinic V${}_{6}$O${}_{13}$ in 2 with zone axis parallel to [100] direction in each lattice. (**f**) nanobeam diffraction from the area 3 in the sample’s surface; the phase is concluded to be monoclinic VO${}_{2}$. (**g**) enlarged fragment of (**c**) indicated by the green dashed square; A thin well-defined strip is separated by crystalline V${}_{2}$O${}_{5}$. The FFT inset—corresponding to the thin strip—indicates it consists of monoclinic VO${}_{2}$ and some amorphous material.

**Table 1 materials-15-07652-t001:** Comparison of this work to other studies investigating a Joule heating-induced resistive switch in VO${}_{2}$. Note in this work the low *R* and the correlated exceptionally low $V_{th}$/*d*. $V_{th}$ in this work is comparable to others due to the large *d*.

	*d* ^1^	$V_{\mathrm{th}}$ (V) ^2^	$V_{\mathrm{th}}$/d (V/m)	*R* (Ω) ^3^
This Work	2 mm	0.85	425	200
Mun et al. [6]	0.2 mm	6.8	34,000	2 × 10${}^{4}$
Yoon et al. [5]	5 μm	14	2.8 × 10${}^{6}$	2 × 10${}^{4}$
Li et al. [8]	6 μm	3.47	1.25 × 10${}^{6}$	8 × 10${}^{4}$
Lin et al. [23]	0.5 μm	0.8	1.6 × 10${}^{6}$	2 × 10${}^{3}$

^1^*d* is the separation between electrodes. ^2^
*V_th_* is taken at temperatures around 1–3 K below the transition temperature. ^3^
*R* corresponds to that of the low-temperature, high-resistance state.

## Data Availability

The data that support the findings of this study are available on request from the corresponding author.

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
