# Peer review of "VOx Phase Mixture of Reduced Single Crystalline V2O5: VO2 Resistive Switching"

_materials, 2022, doi:10.3390/ma15217652_

Round 1

Author Response

Dear Reviewer,

Thank you for your consideration of our work. Below are our responses to your comments.

  1. We have removed the dotted lines from the text. We agree the figure is sufficient.

  2. Indeed, such calculations would support our work. We have experience with DFT calculations of oxides. Unfortunately, binding energy calculations are out of the scope of this work given the time and resources required. Such calculations would require considerable computational power (and hence time) due to its all-electron nature. The majority of DFT calculations make use of pseudo-potentials, which only aim to model the outer electrons, far from the core. This is a suitable for materials properties dictated by the outer electrons around the Fermi level. However, it is not suitable for core level electrons probed by XPS.

    The binding energy positions of the different V oxidation states is experimentally established (reference in text), and hence, we have confidence in our attribution of oxidation states in the two presented XPS spectra.

  3. We have made this change to the vectors in the figure.

  4. We have checked and modified the text where necessary.

Best regards,
Brian Walls and co-authors

Reviewer 2 Report

The present work is targeted to achieve better "metal-to-insulator transition" and is of high relevance. The manuscript is well written, the content is analysed well and the reviewer feels that, it has credibility. The following technical comments may be addressed by the authors while revising their manuscript:

1. What is the cost-effectiveness of the proposed VOx phase mixture? How can this be applied for bulk applications and their potential implications?

2. Explain the possible hysteresis reduction and compare them with existing single crystalline phases that are in vogue while revising the manuscript.

3. What is the coefficient of variation associated with the percolation paths of VO2 and metallic V6O13? How does it compare with existing materials?

4. How does the oxygen flowrate affect the crystallizatin of V2O5 single crystals? How does it practically effect the end performance?

5. Fix the issue in Page 3, Line 92 (Before Reference 24) while revising the manuscript.

6. How can the rate of annealing temperature affect the crytal and eventually the overall efficiency? May be explained in detail while revising the manuscript.

7. The reviewer lauds the efforts of authors for constructively describing the asymmetric and multi phases of vanadium oxide from Lines 100 to 140. It is higly commendable.

8.  Fix the issue in Lines 133 and 134 following each of the references.

9. The caption for Fig 8 can be revised. Most of the information is provided in the figure itself. The authors can give a thought. 

10. Fix the issue with Table 1 heading (Page 10). The information provided following table heading can be provided in the form of a footnote at the end of the table.

Author Response

Dear Reviewer,

Thank you for your consideration of our work. Below are our responses to your comments.

  1. In this study, we examine a V2O5 single crystal. The next step towards application is to perform a similar study with thin films utilizing the knowledge of the presented work. For this reason, I will discuss the cost effectiveness of the thin film growth process and the subsequent annealing required to form the phase mixture. With cost-effectiveness in mind, V2O5 powder or vanadium precursors are the best options. V is an abundant metal and oxidises in the ambient to form V2O5. The relatively cheap powder can be utilized as a target for thin film growth.
    We note that vanadium oxide thin films can be synthesized at low-temperatures [10.1021/acsami.7b03398 & 10.1016/j.matlet.2010.08.022]. To produce the phase mixture one must vacuum anneal a V2O5 film. This requires temperatures around 800K for a short time period (under an hour) in the case of a thin film. This is only aspect of the process which is not cost-effective. However, as a whole, the process is cost-effective.

  2. We realize the resistance vs temperature data presented in Figure 3 was not discussed in the text. The following text has been added to page 6;

    Figure 3 depicts a resistivity measurement of the pristine and annealed crystal in the temperature range 325-380K with the values normalised to the V2O5 resistivity at 325K. The resistivity of a pristine crystal at 325K has been calculated to be (3.0±0.7)x103Ωcm, comparable to literature. Focusing on the annealed crystal, there is a clear drop in resistance at 342K characteristic of the VO2 MIT. This transition sees a reduction in the resistance by a factor of 20 across 5K. A thermal hysteresis of 2K is present, in comparison to the 1K hysteresis observed in the case of single crystalline VO2 [3]. The VO2 is concluded to exhibit only a small range of stoichiometry around dioxide stoichiometry due to the narrow width and hysteresis of the transition.

    We thank the review for bringing this to our attention.

  3. The nature of percolation paths in terms structure (geometric and elemental) and their variation between different switching events is – in this author’s opinion – an important question. The variation between percolation paths can dictate variation in switching characteristics and device failure, for example. This is a difficult question to explore due to the finite size of the percolation path, which requires localized techniques to probe. In addition, one must switch the VO2 via thermal or electrical stimulus during the measurement. We plan to examine the percolation path and its variation via transmission electron microscopy. In principle, the sample can be heated and the change in conductivity can be examined by electron energy loss spectroscopy (EELS), which measures the energy gap of the different vanadium oxide phases correlated to conductivity. This work is not in the scope of the presented work, as it requires careful planning and approved measurement time at a dedicated facility. Objectively, it will be reported at a later date.

    In paragraph 3 of the Discussion, a study which examined the size of the percolation path as a function of device geometry by optical microscopy is detailed.

  4. In this work, we study the reduction of a V2O5 single crystal. We have not considered different vanadium-oxide single crystals. Starting with a lower oxide single crystal will certainly affect the reduction (or oxidation) process, and hence, the resistive switching. Starting with, for example, a VO2 single crystal and oxidising it, will very likely not result in the same mixed phase microstructure as that depicted in the TEM, as this is specific to the reduction of V2O5. The resistive switching is strongly correlated to the structure of the phase mixture in the surface region, and hence, starting with a different VOx crystal will certainly affect the end performance.

  5. We have changed the notation from {} to <> throughout. Thank you for bringing this to our attention.

  6. We have discussed this in the supplemental information. In the main text, we present a study of a crystal annealed at one specific condition (800K for 16h in high vacuum). We have annealed different crystals at different temperatures for varying durations. The temperatures were 600, 700, 800 and 900K. In SI2 left panel we present the resistance-vs-temperature data for each of the temperature anneals. Clearly, 800K presents the largest VO2 resistance change at the transition and the lowest resistance, which is critical for the low-voltage resistive switching.

    In addition, the right panel of SI2 presents the evolution of the resistance-vs-temperature as a function of duration of anneal at 800K. Evidently, the magnitude of the resistance change at the transition increases with duration of anneal, correlated to the weight percent of VO2. Further duration of annealing does not significantly alter the resistance-vs-temperature.

    In terms of overall efficiency or applicability towards low voltage switching of VO2, the ideal characteristics are a VO2 transition and a low resistance of the lower-temperatures, high resistance state. These two characteristics are at odds with each other; lowering the resistance of the lower-temperature, high resistance state reduces the magnitude of the transition. It is clear that 800K represents the best annealing temperature of those examined.

    Returning to our first point. The main text examines the most favourable annealing condition. The text characterises the crystal, the resistive switching and its mechanism, in depth. We are continuing our work examining the reduction of the V2O5 single crystal under different reducing conditions, and endeavour to present a separate comprehensive study on that topic. Much of the work is presented in the supplemental information, however, a TEM study is planned to provide further insight into the reduction process as a function of the reducing conditions.

  7. We thank the reviewer for this comment.

  8. This is adressed in comment 5.

  9. We assume this is a typo as there is no figure 8. We are happy to consider changes if the reviewer can clarify.

  10. We have made this change.

Best regards,
Brian Walls and co-authors

Round 2

Reviewer 1 Report

Dear editor,

I think the manuscript was much improved, and I suggest accepting it.  Think you.

Regards,

ZR